# Effects of an Appeasing Substance Application at Weaning on Growth, Stress, Behavior, and Response to Vaccination of *Bos indicus* Calves

**DOI:** 10.3390/ani13193033

**Published:** 2023-09-27

**Authors:** Douglas Gomes Vieira, Marcelo Vedovatto, Henrique Jorge Fernandes, Eduardo de Assis Lima, Marcella Candia D’Oliveira, Uriel de Almeida Curcio, Juliana Ranches, Matheus Fellipe Ferreira, Osvaldo Alex de Sousa, Bruno Ieda Cappellozza, Gumercindo Loriano Franco

**Affiliations:** 1Faculdade de Medicina Veterinária e Zootecnia, Universidade Federal de Mato Grosso do Sul, Campo Grande 79070-900, MS, Brazil; douglas10dgv@gmail.com (D.G.V.); eduardo.assis@ufms.br (E.d.A.L.); marcellacandia@gmail.com (M.C.D.); urielcurcio@gmail.com (U.d.A.C.); gumercindo.franco@ufms.br (G.L.F.); 2Dean Lee Research and Extension Center, Louisiana State University, Alexandria, LA 71302, USA; 3Unidade Universitaria de Aquidauana, Universidade Estadual de Mato Grosso do Sul, Aquidauana 79200-000, MS, Brazil; henrique.uems@hotmail.com; 4Eastern Oregon Agricultural Research Center, Oregon State University, Burns, OR 97720, USA; juliana.ranches@oregonstate.edu (J.R.); ferreirm@oregonstate.edu (M.F.F.); 5Nutricorp, Araras 13601-000, SP, Brazil; desousaoa@gmail.com; 6Chr. Hansen, 2970 Hørsholm, Denmark; brbrie@chr-hansen.com

**Keywords:** cortisol, immune system, pheromone, temperament

## Abstract

**Simple Summary:**

Weaning is one of the most stressful experiences for beef calves during their life cycle; however, it is a crucial practice for the production system. Calming pheromones can alleviate stress and reduce its negative impacts on behavior, growth, and the immune system. In this study, the administration of a bovine-appeasing substance enhanced body weight gain, reduced temperament scores and serum cortisol concentration, and improved the behavior and response to vaccination. Therefore, the bovine-appeasing substance could be used in complement to the best weaning practices to reduce stress and its impact on performance, temperament, grazing behavior, and the immune system.

**Abstract:**

An analog of a bovine-appeasing substance (BAS) was previously demonstrated to have calming effects, and it could be an alternative to alleviate the stress caused by weaning. Thus, the objective of this study was to evaluate the effects of BAS administration at weaning on growth, stress, behavior, and response to vaccination of Nellore calves. Eighty-six Nellore calves (40 females and 46 males) were abruptly weaned and randomly assigned into 1 of 2 treatments: (1) saline solution (0.9% NaCl; *n* = 43) and (2) BAS (Secure Catte, IRSEA Group, Quartier Salignan, France; *n* = 43). The solutions were topically applied (5 mL/calf) to the nuchal skin area of each animal. On d 0, before treatment application, calves were vaccinated against infectious bovine rhinotracheitis (IBR), parainfluenza-3 (PI_3_) virus, and bovine viral diarrhea virus types 1 and 2 (BVDV-1 and 2). Calves from each treatment were kept in different pastures for 15 d (time of BAS action) and then moved to a single pasture. Body weight (BW), blood samples, and temperament in the chute (entry score, chute score, and exit score) were collected on d 0, 3, 8, 15, 51, and 100, and behavior on pasture on d 1, 2, 4, 5, 6, 7, and 9. Calves assigned to BAS vs. Saline treatment tended to have greater BW on d 15 (*p* = 0.10), tended to have lower entry scores on d 8 and 51 (*p* = 0.10), and chute scores on d 8 (*p* = 0.07), and had lower exit scores on d 8 (*p* = 0.02). Calves assigned to BAS vs. Saline treatment also had greater time grazing on d 7 and 9 (*p* < 0.01), eating concentrate on d 2, 5, and 6 (*p* = 0.05), walking on d 1, 2, 5, and 9 (*p* < 0.01), standing and ruminating on d 2, 7 and 9 (*p* < 0.01), and playing on d 2, 4, 6, 7, and 9 (*p* < 0.01). Furthermore, they had lower time lying on 1 and 2 (*p* < 0.01), standing on d 5 and 9 (*p* < 0.01), and vocalizing on d 1 and 2 (*p* < 0.01). Calves assigned to BAS vs. Saline treatment had greater serum titter concentrations of PI_3_ t on d 15 and 51 (*p* = 0.05) and BVDV-1 on d 51 (*p* = 0.02). However, they had lower serum concentrations of cortisol on d 3 (*p* = 0.03). BAS administration did not affect (*p* ≥ 0.12) the serum titer concentration of IBR and BVDV-2 titers or the plasma concentration of haptoglobin and ceruloplasmin. The BAS administration improved BW, reduced temperament and serum cortisol concentration, and improved behavior and response to vaccination.

## 1. Introduction

Weaning is a highly stressful event for calves, exposing them to many physiological challenges that directly impact their performance, health, and welfare [1]. Upon weaning, calves experience multiple stressors, such as maternal separation and changes in the social and physical environment. Additionally, handling management at weaning is often used as a convenient time for vaccination, deworming, dehorning, castration, branding, and frequently commingling with a new group [2]. The encounter with such stressors can increase plasma cortisol and acute phase proteins and reduce feed and water intake, growth, and the immune system [3].

The behavior of calves is drastically affected for approximately one week after weaning [4]. The most characteristic behavior affected is an increased frequency of vocalization, followed by increased time walking and standing and decreased time grazing, ruminating, playing, and resting [4]. Furthermore, increased aggressivity is frequently observed on pasture or in the corral after weaning [4].

One of the current strategies designed to reduce stress at weaning is using appeasing pheromones, as demonstrated in several species [5,6,7]. In cattle, an analog of a bovine-appeasing substance (BAS) was developed based on a mixture of fatty acids representing the composition of the natural substance produced by the mammary gland [8,9]. The BAS is topically applied to the nuchal skin area of each animal, inducing the calves to inhale the product, which leads to calming effects for about 15 d, according to the manufacturer (Secure Cattle, IRSEA Group, Quartier Salignan, France).

When the BAS was applied at weaning to grazing beef calves [9,10] or kept in a feedlot [11,12], it increased the subsequent growth [9,10], reduced the plasma concentration of haptoglobin [10], concentration of cortisol on hair [12] and plasma [11], aggressivity [12], increased the plasma concentration of glucose and β-hydroxybutyrate [11], and improved the response to vaccination [12].

Although some studies have evaluated the application of BAS in calves [9,10,11,12], we are unaware of a study that evaluated the effects of the application of BAS at weaning on the temperament, behavior, and immune response of grazing *Bos indicus* beef calves. We hypothesized the BAS application at weaning would reduce the stress as observed in other studies [9,10,11,12], and as stress affects performance, behavior, and the immune system [3], we expected BAS application would improve growth, temperament in the corral, diurnal pasture behavior, and humoral immunity and reduce the acute phase response related to stress. Thus, the objective of this study was to evaluate the effects of BAS application at weaning on growth, stress, behavior, and response to vaccination of Nellore calves.

## 2. Materials and Methods

### 2.1. Animals, Treatments, and Sample Collection

The experiment was conducted at the farm school at Universidade Federal do Mato Grosso do Sul in Terenos, MS, Brazil (20°26′50.8″ S 54°50′21.5″ W) during the dry season period, from May to August of 2021.

Eighty-six Nellore calves [40 females and 46 males; the average body weight (BW) was 198 ± 30.8 kg, and the average age was 8 ± 1 month] were abruptly weaned (d 0) and enrolled in the experiment. On d 0, calves were stratified by BW and sex and randomly assigned into 1 of 2 treatments: (1) saline solution (0.9% NaCl; *n* = 43; 20 females and 23 males) and (2) BAS (Secure Cattle, IRSEA Group, Quartier Salignan, France; *n* = 43; 20 females and 23 males). The BAS composition consists of a mixture of fatty acids, including palmitic, oleic, and linoleic acids (similar to the BAS produced by the mammary gland of the cow), added at 1% of the excipient and estimated to remain in treated animals for 15 d, according to the manufacturer [12]. The solutions were topically applied (5 mL/calf) to the nuchal skin area of each animal. Before the treatment applications, calves were segregated by treatment into two groups. Saline-treated calves were processed and immediately released to a pasture before BAS administration in the other group. This treatment application method was chosen to avoid any cross-effects of BAS on Saline-treated calves.

On d 0, before treatment application, calves were vaccinated against infectious bovine rhinotracheitis (IBR), parainfluenza-3 (PI_3_) virus, bovine viral diarrhea virus type 1 and 2 (BVDV-1 and 2), bovine respiratory syncytial virus, and *Mannheimia haemolytica* (2 mL s.c.; Bovi Shield Gold One Shot, Zoetis, São Paulo, SP, Brazil). From d 0 to d 15, calves were maintained in two similar 2-ha pastures (1 pasture/treatment) of marandu-grass [*Urochloa brizantha* (Hochst. ex A. Rich) R. D. Webster, cv. Marandu] complemented with hay [*Urochloa dictyoneura* (Fig. and De Not.) Veldkamp]. Calves from each treatment were kept in different pastures for 15 d to avoid the cross-effects of BAS on Saline-treated calves. On d 15, calves were moved to a single 17-ha pasture (both treatments in the same pasture) of marandu grass, where they were kept until the end of the study (d 100). During the entire experiment, calves had free-choice access to water and protein concentrated supplement [guarantee levels of 300 g/kg of crude protein (CP), 246 g/kg of non-nitrogen protein, 75 g/kg of sodium, 45.2–52.8 g/kg of calcium, 7.30 g/kg of phosphorus, 7.54 g/kg of sulfur, 200 mg/kg of copper, 133 mg/kg of manganese, 600 mg/kg of zinc, 12 mg/kg of cobalt, 15 mg/kg of iodine, 3.6 mg/kg of selenium, and 160 mg/kg of monensin; target intake of 1 g/kg of BW; Criatec 30 Seca, Tec Agro Nutrição Animal, Campo Grande, MS, Brazil).

The BW of calves was collected individually at 08:00 a.m. on d 0, 8, 15, 51, and 100. Fasted BW was not obtained to avoid shrink-induced stress effects on blood parameters evaluated in the study [13]. Three trained technicians evaluated the temperament (same trained personnel during the entire experiment, who were blinded to treatments) in the corral on d 0, 3, 8, 15, 51, and 100. The entry and exit scores in the squeeze chute were evaluated according to Baszczak et al. [14], with scores 1 = animals that walked in or out of the chute; 2 = those that trotted to or from the chute; and 3 = those that ran or galloped in or out of the chute. The chute score was evaluated using an adaptation of Cooke et al. [15], where 1 = calm with no movement; 2 = restless movements; 3 = frequent movement; 4 = constant movement, vocalization, and shaking of the chute; and 5 = violent and continuous struggling.

After the handling in the squeeze chute, calves were sequentially allocated in a small pen (5 × 10 m) and in small groups (*n* = 5) to evaluate the pen score as described by Hammond et al. [16]. Briefly, one trained technician approached the group and evaluated the response of each calf, being scores 1 = nonaggressive (docile)—walks slowly, can approach closely, not excited by technician or facilities; 2 = slightly aggressive—runs along fences, will stand in the corner if humans stay away, may pace fence; 3 = moderately aggressive—runs along fences, head up and will run if humans move closer, stops before hitting gates and fences, avoids humans; 4 = aggressive—runs, stays in the back of group, head high and very aware of humans, may run into fences and gates even with some distance, will likely run into fences if alone in pen; and 5 = very aggressive—excited, runs into fences, runs over humans and anything else in path, “crazy”. Temperament scores were averaged among technicians.

Blood samples were collected from a jugular vein (14 calves/treatment; 7 females and 7 males/treatment) on d 0, 3, 8, 15, 51, and 100 into two blood collection tubes (10 mL; Vacutainer, Becton Dickinson, Franklin Lakes, NJ, USA) with and without sodium heparin for the collection of plasma and serum, respectively. After collection, blood samples were immediately stored on ice and then centrifuged at 1200× *g* for 30 min for plasma and serum harvest. Samples were stored at −20 °C for further analysis of serum concentrations of cortisol and antibody titters (against IBR, PI_3,_ and BVDV-1 and 2) and plasma concentrations of haptoglobin and ceruloplasmin. Cortisol, ceruloplasmin, and haptoglobin were analyzed on d 0, 3, 8, 15, 51, and 100, and antibody titers were analyzed on d 0, 15, and 51. On d 0, calves were individually identified on both sides of the body, with large numbers, using hair dye to facilitate animal identification for behavior evaluation. The diurnal behavior was evaluated from 06:30 a.m. to 06:00 p.m. (678 min/d) on d 1, 2, 4, 5, 6, 7, and 9. An observation tower (elevated 5 m from the ground) was used to facilitate the visualization of the animals by the evaluators on pastures. The tower was built of metal, and the top had a 1.30-meter-high base that hid most of the evaluator’s bodies from the visualization by the calves. A pair of evaluators (one using binoculars and the other recording the behavior in the spreadsheet; each pair was replaced after four hours of evaluation) scanned the calf’s behavior at each 10-min interval, and the specific behavior detected during each scan was counted for 10 min for the statistics analysis. The variables evaluated in each scan were adapted from Enríquez et al. [4], being grazing, eating concentrate, drinking water, walking, lying, lying ruminating, standing, standing ruminating, playing (jumping, running, with no sign of stress), and vocalizing. The time spent on each activity was calculated as a percent of the total time evaluated per day (678 min) for the subsequent statistical analysis.

The herbage mass was evaluated on d 0, 15, 51, and 100 using the comparative yield method [17], and the samples collected were dried at 60 °C for 5 d and weighed. Herbage allowance was calculated as the average herbage mass divided by the average total BW of calves in each pasture [18]. Hand-picked forage samples were also collected on d 0, 15, 51, and 100. Afterward, samples were dried at 60 °C for 5 d and ground at 1 mm for later chemical composition analysis.

### 2.2. Laboratory Analysis

Antibody titers against IBR, PI_3_, and BVDV-1 and 2 viruses were assessed using procedures outlined by Rosenbaum et al. [19]. Individual serum samples were evaluated for the greatest dilution of antibody titers that achieved total protection of cells against those viruses and are reported as log_2_. Calves with antibody titers ≥ 4 for each virus were considered seropositive and assigned a value of 1, whereas calves with antibody titers < 4 were considered seronegative and assigned a value of 0. These scores were utilized to determine the percentage of calves that had positive seroconversion for antibody protection against those viruses, as previously described by Richeson et al. [20].

Plasma concentrations of haptoglobin were analyzed as described by Cooke and Arthington [21] and ceruloplasmin as described by Demetriou et al. [22]. The inter- and intra-assay CV was 3.9% and 6.4% for haptoglobin and 2.0% and 4.3% for ceruloplasmin, respectively. The serum concentration of cortisol was analyzed (Immulite 1000; Siemens Medical Solutions Diagnostics, Los Angeles, CA, USA) as previously described by Cooke et al. [23] due to 100% cross-reactivity between bovine and human cortisol and accomplished within a single assay with an intra-assay with a CV of 8.52%.

Forage samples were analyzed according to AOAC [24]: CP, method 976.05; ether extract (EE), method 920.39; and ash, method 942.05. The concentrations of lignin, neutral detergent fiber (NDF), and acid (ADF) were analyzed as described by Van Soest et al. [25]. The total digestible nutrients (TDN) concentrations were calculated as described by Weiss et al. [26], and the net energy for maintenance (NEm) and gain (NEg) was determined by the equations proposed by the NASEM [27]. The chemical compositions of forage and hay are described in Table 1.

### 2.3. Statistical Analysis

The calf was considered the experimental unit for all analyses. All continuous variables were analyzed using the MIXED procedure, and binomial variables (only the percentual of seroconversion against IBR, PI_3_, and BVDV-1 and 2) were analyzed using the GLIMMIX procedure of SAS (SAS Inst. Inc., Cary, NC, USA; version SAS University), with Satterthwaite approximation to determine the denominator degrees of freedom for the test of fixed effects. All variables were analyzed as repeated measures, and the statistical model used was:*Yijk* = *μ* + *Ti* + *Dj* + *Ak* + *Sl* + (*TD*)*ij* + *eijkl*
where *Yijkl* = observation of the effect of treatment *i* per days of collection *j* in animal *k* and sex *l*; *μ* = overall mean; *Ti* (fixed) = effect of treatment [*i* = 1 (Saline) and 2 (BAS)]; *Dj* (fixed) = effect of days (j = 0, …, 100); *Ak* (random) = animal effect (k = 1, …, 86); *Sl* (random) = sex effect [*l* = 1 (male) and 2 (female)]; *TDij* (fixed) = interaction between treatment *i* and day *j*; and *eijkl* = random error associated with each observation.

The results of day 0 for BW, temperament, and plasma and serum variables were included as covariates in each respective analysis but were removed from the model when *p* > 0.10. The Toeplitz covariance structure was selected for BW, and the first-order autoregressive covariance structure was selected for all other variables. The covariance structures were selected according to the lowest Akaike information criterion. Means were separated using pairwise differences (PDIFF), and all results were reported as the least squares mean (LSMEANS), followed by the standard error of the mean (SEM). Significance was defined as *p* ≤ 0.05, and tendency was defined as *p* > 0.05 and ≤0.10. The Pearson correlations were analyzed using the CORR procedure of SAS (to evaluate correlations between BW, temperament, and blood variables).

## 3. Results

A tendency for effect of treatment × day interaction was detected (*p* = 0.10) for BW, in which BAS-treated calves had greater BW on d 15 than Saline-treated calves (Table 2). An interaction between treatment × day was detected for average daily gain (ADG; *p* < 0.01), where BAS-treated calves showed greater ADG from d 8 to 15 compared to Saline-treated calves (Table 2).

Upon treatment administration, an effect of treatment (*p* ≤ 0.05) and a tendency for treatment × day interaction were detected (*p* = 0.10) for the entry score, in which BAS-treated calves had lower scores on d 8 and 51 compared to Saline-treated calves (Table 3). Furthermore, a tendency for treatment × day effect (*p* = 0.07) was detected for chute score (*p* = 0.02) and exit score, being lower for BAS-treated calves only on d 8, compared to Saline-treated calves (Table 3). No effects of treatment or treatment × day interaction (*p* ≥ 0.11) were detected for the pen score (Table 3).

The diurnal behavior was affected by treatments, and a treatment × day interaction was detected in several variables. The BAS-treated calves spent more time grazing (*p* < 0.01; on d 7 and 9), eating concentrate (*p* = 0.05; on d 2, 5 and 6), walking (*p* < 0.01; on d 1, 2, 5 and 9), and standing ruminating (*p* < 0.01; on d 2, 7 and 9) and less time lying (*p* < 0.01; on d 1 and 2) and standing (*p* < 0.01; on d 5 and 9) compared to Saline-treated calves (Table 4). No effects of treatment or its interaction with days (*p* ≥ 0.18) were detected for drinking water and lying and ruminating (Table 4). Furthermore, BAS-treated calves spent more time playing (*p* < 0.01; on d 2, 4, 6, 7, and 9) and less time vocalizing (*p* < 0.01; on d 1, 2, and 3) compared to Saline-treated calves (Figure 1 and Figure 2).

The response to vaccination was affected by treatments, and a treatment × day interaction was detected (*p* = 0.05) for PI_3_ titters, being greater for BAS-treated calves on d 15 and 51 compared to Saline-treated calves (Table 5). A treatment × day interaction also tended to be detected (*p* = 0.10) for PI_3_ seroconversion, being greater for BAS-treated calves on d 15, compared to Saline-treated calves. Furthermore, a treatment × day interaction was detected (*p* = 0.02) for BVDV-1 titters, being greater for BAS-treated calves on d 51 compared to Saline-treated calves. No effects of treatment × day (*p* = 0.69) were detected, but treatment effects tended to be detected (*p* = 0.09) for BVDV-1 seroconversion, being greater for BAS-treated calves than Saline-treated calves. No effects of treatment or its interaction with days (*p* ≥ 0.12) were detected for titers and seroconversion of IBR and BVDV-2 (Table 5).

The plasma concentration of haptoglobin and ceruloplasmin was not affected by treatment × day or treatment (*p* ≥ 0.36; Table 5). However, a treatment × day interaction was detected (*p* = 0.03) for serum cortisol concentration, which was lower for BAS-treated calves on d 3, compared to Saline-treated calves (Figure 3).

Significative Pearson correlations were detected between several experiment variables (*p* ≤ 0.05; Figure 4). The BW had positive correlations only with plasma ceruloplasmin concentration. Temperament variables (entry, chute, exit, and pen scores) were all positively correlated, as expected, and temperament variables had negative correlations with IBR and BVDV-1 titter concentrations but had positive correlations with serum cortisol concentration. However, BVDV-2 titter concentration was positively correlated with serum cortisol concentration. No correlations between temperament variables and plasma haptoglobin and ceruloplasmin were detected (*p* > 0.05).

## 4. Discussion

Weaning is a highly stressful event for calves, exposing them to physiological, physical, and psychological challenges that directly impact their performance, health, and welfare [1]. In the current experiment, BAS application was able to improve BW, reduce stress, and improve behavior and immune response immediately after weaning, corroborating the immediate window of efficacy of BAS (15 d).

In the current experiment, the BAS-treated calves had greater BW only on d 15, compared to the Saline-treated calves. Other studies also observed improved ADG in calves when BAS was applied at weaning [9,10,11]. The absence of BAS effects after d 15 is probably related to the time of action of the product, which is about 15 d, according to the manufacturer. The improved ADG in BAS-treated calves was probably a consequence of the reduced stress response (i.e., lower temperament scores, lower serum concentration of cortisol, and lower vocalization) and its effects on behavior (mainly by increasing time spent grazing and consuming supplements).

In the current study, the animals did not gain BW from the beginning to the end of the experiment (d 0 to 100). This is explained by the low quality of the forage/hay and the low herbage allowance during the study, which is a common scenario in tropical regions. Although a protein supplement was offered to calves post-weaning to attenuate the low crude protein concentrations in forage and hay, it was not enough to promote a continuous gain post-weaning. This scenario increased the stress of the experimental calves; however, all animals from both treatments were subjected to the same environment and stress conditions.

The temperament scores and the serum concentration of cortisol were reduced by BAS application, indicating less stress on those calves, mainly during the first two weeks after weaning. In a study conducted in a feedlot with crossbreed (90% British × 10% Nellore) weaned calves, BAS administration reduced the exit velocity by 14 d after weaning, and it also reduced the hair concentration of cortisol on d 14 [12]. In another study with weaned Angus-influenced calves raised in a feedlot, the BAS administration at weaning reduced the plasma concentration of cortisol and increased the plasma concentration of glucose 7 d after weaning [11]. In addition to those results from previous studies with *Bos taurus*-influenced calves raised in a feedlot, we are unaware of a study that evaluated the BAS application on temperament and serum concentration of cortisol in *Bos indicus* calves raised in pasture. Our study demonstrates that BAS administration can reduce the adrenocortical response elicited by weaning, and this reduction can affect calf temperament and BW.

The mechanism through which BAS administration leads to reduced cortisol production is still unclear. However, it is known that BAS target organs involved in pheromone perception include the main olfactory epithelium (MOE) and vomeronasal organ (VNO; [28,29]). The MOE is responsible for the recognition of traditional odor molecules and chemical and environmental signals without specificity or meaning, whereas the VNO is related to pheromone recognition, carrying specific chemosensory signals through the receptors [30], leading to the occurrence of a neuroendocrine cascade [29]. The VNO neurons can encode stimulus strength, activating an entire neural subpopulation and conducting an electrochemical signal to the calf brain [28], stimulating the hypothalamus to exhibit an appropriate neuroendocrine response unique to the specific subpopulation of neurons stimulated in the VNO and causing calming effects in the animal [29].

An increase in vocalization and a decrease in time playing are traditional characteristics of behavior presented by calves after weaning due to the high psychological stress caused by the separation from the dam [4]. In the current study, the BAS administration assuaged these effects, which corroborates with the results of temperament and cortisol. In another study, the BAS application reduced the number of escape attempts and mounts on subsequent days after weaning, showing the fast-calming effect of the product on the animal after administration [12].

In the current study, the BAS application increased the time spent grazing, eating concentrate, walking, and standing ruminating and reduced the time spent lying. The effects of BAS on physical activity were also evaluated by Schubach et al. [12], but using a pedometer. They found that BAS-treated calves tended to engage in more allogrooming bouts and presented more steps after weaning. All these results show that BAS-treated calves were more active after weaning, and the increased walking in our experiment was more related to exploratory activity than a greater stress response. Increased time spent grazing, eating concentrate, and ruminating in BAS-treated calves shows faster adaptation to the new environment due to the lower stress. These changes in behavior could also be responsible for the improved BW in BAS-treated calves on d 15 of the study. The lower serum cortisol concentration detected on d 3 in BAS vs. Saline-treated calves was probably responsible for causing these behavior changes between treatments.

Weaning stress-related responses are well known to have a prominent impact on the immune system, leading to immunosuppression in calves [29]. In the current experiment, the BAS application increased the response to vaccination, showing greater humoral immunity against PI_3_ and BVDV-1. Another study with *Bos taurus*-influenced calves also observed better responses to vaccination after BAS application [12]. Cortisol affects the immune system in several ways, including reducing immune cell proliferation and differentiation, effector cell function, and increasing cytokine expression [31]. In our correlation analysis, serum cortisol concentration was negatively correlated with the serum concentration of PI_3_ and BVDV-1 titers, showing the negative effects of cortisol on the immune system. However, surprisingly, cortisol was positively correlated with BVDV-2 titer concentration. The reason for that is unknown and deserves further investigation. However, it could explain the lack of BAS effects on the BVDV-2 titer concentration.

Acute-phase proteins also affect the immune system and are normally increased by stress [29], but in the current study, haptoglobin and ceruloplasmin were not affected by BAS application. The effects of BAS on acute-phase proteins are inconsistent. Some studies have shown decreased concentrations of haptoglobin [9,12], while others have found no differences [11,32]. In our study, the serum cortisol concentration was reduced by BAS only on d 3, and cortisol has been reported to trigger an acute, transient, and temporary inflammatory cascade that traditionally happens approximately 48 to 72 h after the cortisol peak [29]. Based on that rationale, if the cortisol peak was on d 3, the inflammatory response should have happened after d 3, and in this case, the inflammatory reaction could have happened between d 3 and 8. If our rationale is correct, we could have lost the timing of blood collection for ceruloplasmin and haptoglobin analysis (blood collection was made on d 0, 3, 8, 15, 51, and 51), and further studies with less spaced-out blood collection deserve to be conducted.

## 5. Conclusions

The administration of a bovine-appeasing substance at weaning resulted in lower serum cortisol concentrations, concomitant with lower temperament scores, while at the chute, calves consequently engaged in desirable behaviors immediately post-weaning, such as grazing, eating, and rumination, which ultimately led to enhanced body weight gain and response to vaccination. The bovine-appeasing substance could be used in complement to the best weaning practices to reduce stress and its impact on performance, temperament, pasture behavior, and the immune system.

## Figures and Tables

**Figure 1 animals-13-03033-f001:**
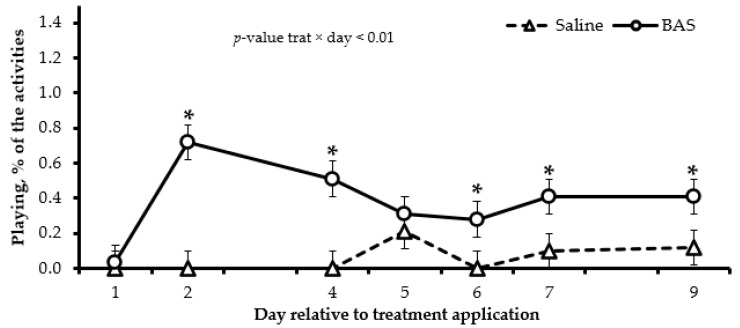
Playing behaviors of Nellore calves receiving saline solution (Saline; *n* = 14) or bovine-appeasing substance (BAS; *n* = 14) at weaning. Total time evaluated = 678 min/day. * Represents differences (*p* ≤ 0.05) between treatments on each day.

**Figure 2 animals-13-03033-f002:**
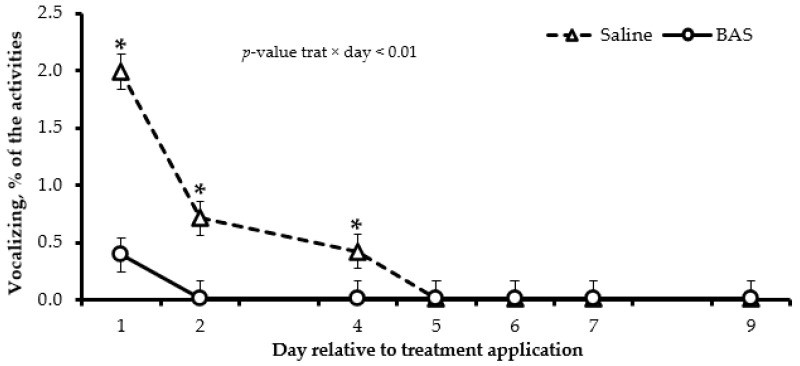
Vocalizing behaviors of Nellore calves receiving saline solution (Saline; *n* = 14) or bovine-appeasing substance (BAS; *n* = 14) at weaning. Total time evaluated = 678 min/day. * Represents differences (*p* ≤ 0.05) between treatments on each day.

**Figure 3 animals-13-03033-f003:**
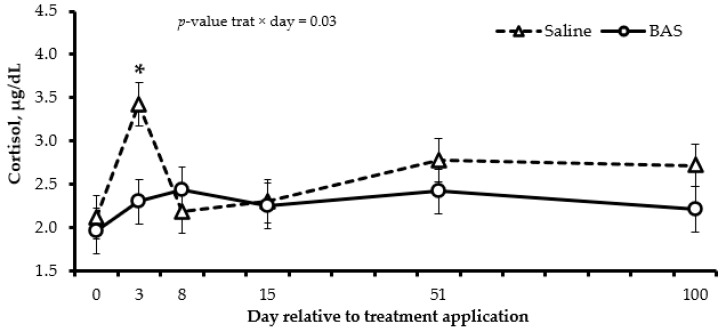
Serum cortisol concentration of Nellore calves receiving saline solution (Saline; *n* = 14) or bovine-appeasing substance (BAS; *n* = 14) at weaning. * Represents differences (*p* ≤ 0.05) between treatments on each day.

**Figure 4 animals-13-03033-f004:**
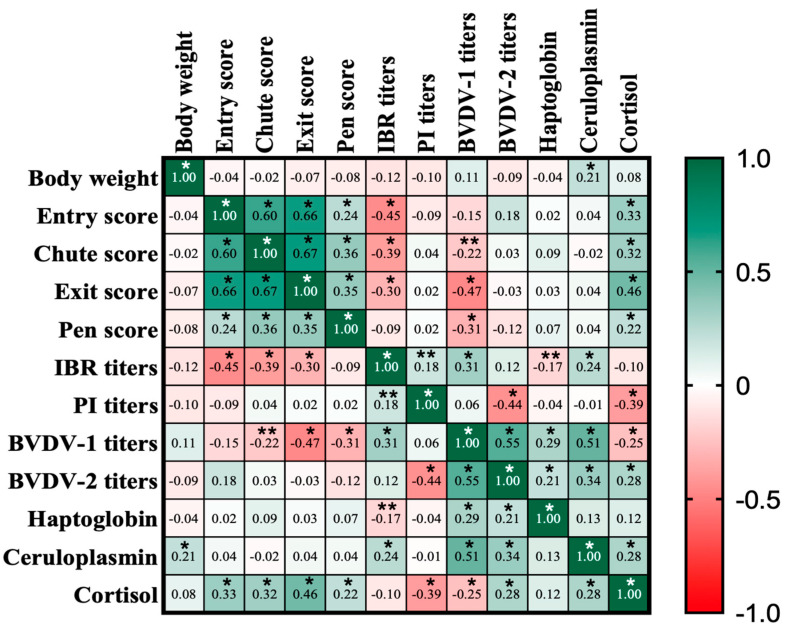
Heat map of Pearson correlations between variables in the experiment. Green colors represent positive correlations, and red colors represent negative correlations. The number in each cell represents the Pearson correlation coefficients. * Represents differences (*p* ≤ 0.05), and ** represents tendencies (*p* ≤ 0.10).

**Table 1 animals-13-03033-t001:** Chemical composition of marundu-grass [*Urochloa brizantha* (Hochst. ex A. Rich) R. D. Webster, cv. Marandu] and dictyoneura-hay [*Urochloa dictyoneura* (Fig. and De Not.) Veldkamp] consumed by calves, herbage mass, and herbage allowance.

Items ^1^	Saline Pasture			BAS Pasture			Saline and BAS in the Same Pasture	
	Forage d0	Forage d15	Hay d0 to d15	Forage d0	Forage d15	Hay d0 to d15	Forage d51	Forage d100
g/kg of DM								
Crude protein	81.0	77.0	101	89.0	80.0	103	91.0	83.0
NDF	622	680	759	620	676	759	645	663
ADF	339	402	403	336	387	404	354	297
Lignin	21.0	45.0	58.0	18.0	45.0	57.0	39.0	31.0
Ether extract	24.0	28.0	19.0	27.0	24.0	19.0	24.0	34.0
Ash	109	118	39.0	110	119	44.0	105	98.0
TDN	649	585	610	659	580	627	612	643
Mcal/kg of DM								
NEm	1.47	1.26	1.34	1.50	1.24	1.40	1.35	1.45
NEg	0.88	0.69	0.77	0.91	0.67	0.82	0.77	0.86
Herbage mass, kg DM/ha	4956	1840		4893	1930		1752	2554
Herbage allowance, kg DM/kg BW	0.56	0.21		0.54	0.22		0.10	0.17

^1^ DM, dry matter; NDF, neutral detergent fiber; ADF, acid detergent fiber; TDN, total digestible nutrients; NEm, net energy for maintenance; NEg, net energy for gain.

**Table 2 animals-13-03033-t002:** Growth performance of Nellore calves receiving saline solution (Saline; *n* = 43) or bovine-appeasing substance (BAS; *n* = 43) at weaning (d 0).

Items	Treatments		SEM	*p*-Value	
	Saline	BAS		Treatment	Treatment × Day
Body weight, kg				0.60	0.10
d 0	198	198	0.96		
d 8	196	196	0.96		
d 15	194 ^b^	197 ^a^	0.96		
d 51	203	203	0.97		
d 100	195	196	0.98		
Average daily gain, kg/d				0.09	<0.01
d 0 to 8	−0.179	−0.268	0.07		
d 8 to 15	−0.278 ^b^	0.158 ^a^	0.07		
d 15 to 51	0.226	0.164	0.07		
d 51 to 100	−0.152	−0.134	0.07		

^a,b^ Within a row, without a common superscript, differ (*p* ≤ 0.05) or tend to differ (*p* ≤ 0.10).

**Table 3 animals-13-03033-t003:** Temperament of Nellore calves receiving saline solution (Saline; *n* = 43) or bovine-appeasing substance (BAS; *n* = 43) at weaning.

Items	Treatments		SEM	*p*-Value	
	Saline	BAS		Treatment	Treatment × Day
Entry score (1–3)				0.05	0.10
d 0	1.59	1.63	0.07		
d 3	1.65	1.50	0.07		
d 8	1.96 ^a^	1.71 ^b^	0.07		
d 15	1.59	1.56	0.07		
d 51	1.87 ^a^	1.69 ^b^	0.07		
d 100	1.48	1.37	0.07		
Chute score (1–5)				0.32	0.07
d 0	2.40	2.52	0.10		
d 3	2.38	2.37	0.10		
d 8	2.52 ^a^	2.21 ^b^	0.10		
d 15	2.04	2.07	0.10		
d 51	2.26	2.06	0.10		
d 100	2.13	1.93	0.10		
Exit score (1–3)				0.10	0.02
d 0	1.87	1.92	0.07		
d 3	1.81	1.77	0.07		
d 8	2.09 ^a^	1.72 ^b^	0.07		
d 15	1.82	1.67	0.07		
d 51	1.94	1.94	0.07		
d 100	1.77	1.65	0.07		
Pen score (1–5)	2.31	2.40	0.05	0.22	0.11

^a,b^ Within a row, without a common superscript, differ (*p* ≤ 0.05) or tend to differ (*p* ≤ 0.10).

**Table 4 animals-13-03033-t004:** Diurnal pasture behavior of Nellore calves receiving saline solution (Saline; *n* = 14) or bovine-appeasing substance (BAS; *n* = 14) at weaning.

Items, % of the Activities	Treatments		SEM	*p*-Value	
	Saline	BAS		Treatment	Treatment × Day
Grazing				0.56	<0.01
d 1	27.9	27.4	1.84		
d 2	28.3	27.8	1.84		
d 4	36.5	35.7	1.84		
d 5	42.1	42.4	1.84		
d 6	43.2	42.1	1.84		
d 7	45.1 ^b^	50.2 ^a^	1.84		
d 9	36.3 ^b^	49.2 ^a^	1.84		
Eating concentrate				<0.01	0.05
d 1	0.30	0.65	0.58		
d 2	1.95 ^b^	3.39 ^a^	0.58		
d 4	1.54	2.14	0.58		
d 5	2.69 ^b^	4.84 ^a^	0.58		
d 6	1.45 ^b^	4.86 ^a^	0.58		
d 7	1.84	2.45	0.58		
d 9	3.16	3.37	0.58		
Drinking water	1.04	0.85	0.18	0.46	0.53
Walking				<0.01	<0.01
d 1	12.6 ^b^	20.9 ^a^	1.07		
d 2	5.34 ^b^	14.8 ^a^	1.07		
d 4	6.95	6.74	1.07		
d 5	6.42 ^b^	9.70 ^a^	1.07		
d 6	6.53	4.96	1.07		
d 7	7.36	5.61	1.07		
d 9	8.27 ^b^	12.9 ^a^	1.07		
Lying				<0.01	<0.01
d 1	21.7 ^a^	6.22 ^b^	1.53		
d 2	30.2 ^a^	17.3 ^b^	1.53		
d 4	30.6	32.8	1.53		
d 5	26.3	22.8	1.53		
d 6	21.2	18.1	1.53		
d 7	16.7	19.7	1.53		
d 9	21.9	19.2	1.53		
Lying ruminating	2.97	3.76	0.41	0.18	0.42
Standing				0.62	<0.01
d 1	29.9	33.9	1.75		
d 2	31.9	28.8	1.75		
d 4	20.1	17.4	1.75		
d 5	24.3 ^a^	12.5 ^b^	1.75		
d 6	21.1	19.9	1.75		
d 7	15.8	18.3	1.75		
d 9	10.4 ^b^	17.5 ^a^	1.75		
Standing ruminating				<0.01	0.01
d 1	1.19	2.06	0.74		
d 2	1.03 ^b^	4.32 ^a^	0.74		
d 4	2.05	1.33	0.74		
d 5	2.80	3.14	0.74		
d 6	3.94	2.89	0.74		
d 7	1.33 ^b^	3.37 ^a^	0.74		
d 9	1.53 ^b^	4.60 ^a^	0.74		

^a,b^ Within a row, without a common superscript, differ (*p* ≤ 0.05) or tend to differ (*p* ≤ 0.10). Total time evaluated = 678 min/day.

**Table 5 animals-13-03033-t005:** Response to vaccination and acute phase response of Nellore calves receiving saline solution (Saline; *n* = 14) or bovine-appeasing substance (BAS; *n* = 14) at weaning.

Items ^1^	Treatments		SEM	*p*-Value	
	Saline	BAS		Treatment	Treatment × Day
Response to vaccination					
IBR					
Titers, log_2_	2.94	3.06	0.53	0.85	0.66
Seroconversion, % total	91.7	100	5.87	0.33	0.33
PI_3_					
Titers, log_2_				0.03	0.05
d 0	0.00	0.00	0.53		
d 15	3.21 ^b^	5.49 ^a^	0.53		
d 51	4.82 ^b^	6.15 ^a^	0.53		
Seroconversion, % total				0.12	0.10
d 0	0.00	0.00	0.09		
d 15	66.7 ^b^	100 ^a^	0.09		
d 51	100	100	0.09		
BVDV-1					
Titers, log_2_				0.10	0.02
d 0	0.26	0.40	0.61		
d 15	0.26	0.57	0.61		
d 51	4.26 ^b^	6.40 ^a^	0.61		
Seroconversion, % total	66.7	100	12.0	0.09	0.69
BVDV-2					
Titers, log_2_	1.06	1.28	0.42	0.72	0.87
Seroconversion, % total	66.7	66.7	12.2	1.00	1.00
Acute phase response					
Haptoglobin, mg/mL	0.41	0.43	0.02	0.46	0.40
Ceruloplasmin, mg/mL	15.7	14.8	0.64	0.36	0.41

^1^ On d 0, calves were vaccinated against infectious bovine rhinotracheitis, bovine viral diarrhea virus type 1 and 2 (BVDV-1 and 2), parainfluenza-3 (PI3) virus, bovine respiratory syncytial virus, and *Mannheimia haemolytica* (2 mL s.c.; Bovi Shield Gold One Shot, Zoetis). ^a,b^ Within a row, without a common superscript, differ (*p* ≤ 0.05) or tend to differ (*p* ≤ 0.10).

## Data Availability

Data are available by email request to the corresponding author with reasonable justification.

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
