# Peer review of "Effects of an Appeasing Substance Application at Weaning on Growth, Stress, Behavior, and Response to Vaccination of Bos indicus Calves"

_animals, 2023, doi:10.3390/ani13193033_

Round 1

Reviewer 1 Report

General comments

Introduction.

In my opinion, the authors did a good job with the Introduction chapter, justifying the need for research into the use of an appeasing substance at weaning on growth, stress, behavior, and response to calves vaccination.

Material and methods

I suggest supplementing the subchapter "Statistical analysis" with a model of analysis performed using the MIXED and GLIMMIX procedures. Please describe in more detail the performed "principal component analysis"

Results

I consider the description of the obtained results as well as the discussion to be generally correct.

Tables  - enter the units in which individual features were measured: kg, g, etc.

The description of the PCA results is very "frugal".

 Detailed comments

Lines 93-94: Notations “=198+-30. 8kg; *+-1 month of age” is in my opinion unclear.

Line: “Full BW” and “at 0800 h” – unclear/incorrect

Line 158: “0630h”, 1800h” – correct – 6:30 A.M., 6:00 P.M.

 Table 1: I propose to abandon point 2 and 3 of the explanations below the table. This information is contained in the descriptive part of the Material and Methods chapter.

 Line 215: LSMEANS - explain the abbreviation used

 Table 2, 3 and 4: I propose to abandon point 1 of the explanations below the table. This information is contained in the descriptive part of the Material and Methods chapter.

 Lines 245- 256: I don't understand the use of the "BAS x Saline" notation by the authors

 Table 4: The table is too large. In my opinion it should be split into 2 tables.

 Lines 270-279: Limit the number of citations to table 5. The authors refer to this table as many as 5 times.

 Lines 270-279: If the authors write that they were higher or lower, it is necessary to specify the reference group. Please formulate sentences on the principle: were greater than ....

 Line 300: Don't start a sentence with an abbreviation.

Author Response

Introduction.

In my opinion, the authors did a good job with the Introduction chapter, justifying the need for research into the use of an appeasing substance at weaning on growth, stress, behavior, and response to calves vaccination.

Authors: thank you. We appreciate your comment.

Material and methods

I suggest supplementing the subchapter "Statistical analysis" with a model of analysis performed using the MIXED and GLIMMIX procedures. Please describe in more detail the performed "principal component analysis"

Authors: no problem. The model was included in this version. In this version, Table 4 was split into one table and 2 figures. So, to reduce the number of figures, the PCA analysis and the figure were removed.

Results

I consider the description of the obtained results as well as the discussion to be generally correct.

Authors: thank you.

Tables  - enter the units in which individual features were measured: kg, g, etc.

Authors: all checked/included.

The description of the PCA results is very "frugal".

Authors: PCA analyses removed.

 Detailed comments

Lines 93-94: Notations “=198+-30. 8kg; *+-1 month of age” is in my opinion unclear.

Authors: adjusted.

Line: “Full BW” and “at 0800 h” – unclear/incorrect

Authors: adjusted.

Line 158: “0630h”, 1800h” – correct – 6:30 A.M., 6:00 P.M.

Authors: adjusted.

Table 1: I propose to abandon point 2 and 3 of the explanations below the table. This information is contained in the descriptive part of the Material and Methods chapter.

Authors: excluded.

 Line 215: LSMEANS - explain the abbreviation used

Authors: adjusted

 Table 2, 3 and 4: I propose to abandon point 1 of the explanations below the table. This information is contained in the descriptive part of the Material and Methods chapter.

Authors: excluded.

 Lines 245- 256: I don't understand the use of the "BAS x Saline" notation by the authors

Authors: agreed. Sentences were rewritten.

 Table 4: The table is too large. In my opinion it should be split into 2 tables.

Authors: In this version, Table 4 was split into one table and 2 figures.

 Lines 270-279: Limit the number of citations to table 5. The authors refer to this table as many as 5 times.

Authors: agreed and adjusted.

 Lines 270-279: If the authors write that they were higher or lower, it is necessary to specify the reference group. Please formulate sentences on the principle: were greater than ....

Authors: agreed. Sentences were rewritten.

 Line 300: Don't start a sentence with an abbreviation

Authors: agreed and adjusted.

Authors: Thank you. The manuscript was improved with your suggestions.

Reviewer 2 Report

This is a very meaningful paper. This study uses an analogous of the bovine-appeasing substance (BAS) to reduce stress in weaned beef calves and its impact on performance, temperament, pasture behavior, and the immune system. However, the paper is needed to be improved. My detailed comments are as follows:

1.      L79-81: The hypothesis is too generic. Please re-write the hypothesis on why you think the BSA is specifically working on the growth, stress, behavior, and response to vaccination.

2.      In the line 97-98, what are the contents of these fatty acids? How to ensure the BSA remain in treated animals for 15 d?

3.      In the line 156-157, can you introduce the observation tower and the method to evaluate the diurnal behavior (grazing, ruminating, et al.) in detail?

4.      In the line 358-374, it's still unclear why the BSA affect the behavior of the calves. Can you give some explanation?

Author Response

This is a very meaningful paper. This study uses an analogous of the bovine-appeasing substance (BAS) to reduce stress in weaned beef calves and its impact on performance, temperament, pasture behavior, and the immune system. However, the paper is needed to be improved. My detailed comments are as follows:

  1. L79-81: The hypothesis is too generic. Please re-write the hypothesis on why you think the BSA is specifically working on the growth, stress, behavior, and response to vaccination.

Authors: agreed, and the hypothesis was rewritten.

  1. In the line 97-98, what are the contents of these fatty acids? How to ensure the BSA remains in treated animals for 15 d?

Authors: The specific concentration of each FA, unfortunately, is not provided by the company. The time of BAS remains on the animals (15 d) was also evaluated and determined by the company. However, in an experiment conducted by Schubach et al. 2020, the hair concentration of cortisol was reduced by 14 d, and the exit velocity from the chute was also reduced by 14 d after BAS application. So, we assume that the statement from the company is right. I included the Schubach citation at the end of this statement in this current version.

  1. In the line 156-157, can you introduce the observation tower and the method to evaluate the diurnal behavior (grazing, ruminating, et al.) in detail?

Authors: Absolutely. It was adjusted in the text.

  1. In the line 358-374, it's still unclear why the BSA affect the behavior of the calves. Can you give some explanation?

Authors: yes. The lower serum cortisol concentration detected on d 3 on BAS vs. Saline-treated calves was probably responsible for causing these behavior changes between treatments. It was described at the end of the paragraph in this version. The way BAS reduces cortisol was explained in the previous paragraph.

Authors: Thank you. The manuscript was improved with your suggestions.

Reviewer 3 Report

General comment: The article Effects of an appeasing substance application at weaning on growth, stress, behavior, and response to vaccination of Bos indicus calves” is an interesting paper concerning the evaluation of the effects of an analogous of the bovine-appeasing substance (BAS) application at weaning on growth, stress, behavior, and response to vaccination of Nellore calves. The argument object of this paper results of outmost importance and the contents addressed in this study are relevant and worthy of further investigations, from both the speculative and the applied points of view. In fact, weaning is one of the most stressful experiences for beef calves during their life cycle; however, it is a crucial practice for the production system and the evaluation of the effects of BAS for validity and reliability of this treatment could be an alternative to alleviate the stress caused by weaning.

The Introduction clearly introduces the studies that have been performed on current strategies designed to reduce stress at weaning by using appeasing pheromones in different species, and it focuses on the need of a study on the evaluation of the effects of the application of BAS at weaning on  temperament, behavior, and immune response of grazing Bos indicus beef calves. The section of Materials and Methods is clear, correctly and extensively presenting the study design and the methodologies adopted. The study design, the variables detected and the statistical analysis are well conceived in order to collect useful data. The study was carried out on eighty-six Nellore calves (40 females 30 and 46 males) that were abruptly weaned and randomly assigned into 1 of 2 treatments: 1) saline solution (0.9% NaCl; n = 43) and 2) BAS (Secure Catte, IRSEA Group, Quartier Salignan, France; n = 43). The solutions were topically applied (5 mL/calf) on the nuchal skin area of each animal. After vaccination against different pathologies and before treatment application, calves from each treatment were kept in different pastures for 15 d (time of BAS action) and then moved to a single pasture. Thereafter, body weight (BW), blood samples, and temperament in the chute (entry score, chute score, and exit score) were collected on d 0, 3, 8, 15, 51, and 100, and behavior on pasture on d 1, 2, 38 4, 5, 6, 7 and 9. Moreover, serum concentration of cortisol and antibody titters (against IBR, PI3, and BVDV-1 and 2) and plasma concentrations of haptoglobin and ceruloplasmin were also analysed at different times. The results showed that the administration of BAS enhanced body weight gain, reduced temperament scores and serum cortisol concentration, and improved the behavior and response to vaccination. The figures and tables are well conceived in order to present the results. Therefore, the Authors concluded that bovine-appeasing substance could be used in complement to the best weaning practices to reduce stress and its impact on performance, temperament, grazing behavior, and the immune system. The discussion of results follows a logical line; it is extensive and clear.  The comments reported in discussion are pertinent to the data achieved. The authors critically examine the data in the light of the state of science highlighted in the introduction. In conclusion, the manuscript is attractive and easy to read and the results obtained are clearly presented.

The English used in the paper is sufficient and some minor mistakes could be reviewed by the Authors.

The current manuscript is acceptable for publication after minor revision.

Title: It is correct.

Simple Summary and Abstract: They are suitable. They clearly identify the interest for this study and its possible relevance. They recap the information contained in the main text without repetitions.

Introduction: The Introduction provides adequate background. This section is concise, and includes specific literature references.

Materials and Methods: This section is clear, correctly and extensively presenting the study design, the methodology adopted and the subsequent evaluation of results.

Results: The results obtained by the authors are logically presented and accompanied by clear tables and figures. They are extensively described and substantially commented in the light of the aim of the study.  

Discussion: The scientific data presented are pertinent to the aim of the study. The discussion of data is well organized and balanced. The authors critically examine the results of data achieved in the light of the state of science highlighted in the introduction and the comments reported in discussion are pertinent. Discussion follows a logical line. The discussion of results is also extensive and clear.  The conclusions are drawn from the data related to the aim of the study. The paper offers the perspective for further study.

References: They are appropriate and present a good up-to-date of items on the argument.

Decision: The current manuscript is acceptable for publication after minor revision.

The English used in the paper is sufficient and some minor mistakes could be reviewed by the Authors.

Author Response

General comment: The article Effects of an appeasing substance application at weaning on growth, stress, behavior, and response to vaccination of Bos indicus calves” is an interesting paper concerning the evaluation of the effects of an analogous of the bovine-appeasing substance (BAS) application at weaning on growth, stress, behavior, and response to vaccination of Nellore calves. The argument object of this paper results of outmost importance and the contents addressed in this study are relevant and worthy of further investigations, from both the speculative and the applied points of view. In fact, weaning is one of the most stressful experiences for beef calves during their life cycle; however, it is a crucial practice for the production system and the evaluation of the effects of BAS for validity and reliability of this treatment could be an alternative to alleviate the stress caused by weaning.

The Introduction clearly introduces the studies that have been performed on current strategies designed to reduce stress at weaning by using appeasing pheromones in different species, and it focuses on the need of a study on the evaluation of the effects of the application of BAS at weaning on  temperament, behavior, and immune response of grazing Bos indicus beef calves. The section of Materials and Methods is clear, correctly and extensively presenting the study design and the methodologies adopted. The study design, the variables detected and the statistical analysis are well conceived in order to collect useful data. The study was carried out on eighty-six Nellore calves (40 females 30 and 46 males) that were abruptly weaned and randomly assigned into 1 of 2 treatments: 1) saline solution (0.9% NaCl; n = 43) and 2) BAS (Secure Catte, IRSEA Group, Quartier Salignan, France; n = 43). The solutions were topically applied (5 mL/calf) on the nuchal skin area of each animal. After vaccination against different pathologies and before treatment application, calves from each treatment were kept in different pastures for 15 d (time of BAS action) and then moved to a single pasture. Thereafter, body weight (BW), blood samples, and temperament in the chute (entry score, chute score, and exit score) were collected on d 0, 3, 8, 15, 51, and 100, and behavior on pasture on d 1, 2, 38 4, 5, 6, 7 and 9. Moreover, serum concentration of cortisol and antibody titters (against IBR, PI3, and BVDV-1 and 2) and plasma concentrations of haptoglobin and ceruloplasmin were also analysed at different times. The results showed that the administration of BAS enhanced body weight gain, reduced temperament scores and serum cortisol concentration, and improved the behavior and response to vaccination. The figures and tables are well conceived in order to present the results. Therefore, the Authors concluded that bovine-appeasing substance could be used in complement to the best weaning practices to reduce stress and its impact on performance, temperament, grazing behavior, and the immune system. The discussion of results follows a logical line; it is extensive and clear.  The comments reported in discussion are pertinent to the data achieved. The authors critically examine the data in the light of the state of science highlighted in the introduction. In conclusion, the manuscript is attractive and easy to read and the results obtained are clearly presented.

The English used in the paper is sufficient and some minor mistakes could be reviewed by the Authors.

Authors: Thank you for your comments. The English was improved in this current version.

The current manuscript is acceptable for publication after minor revision.

Title: It is correct.

Simple Summary and Abstract: They are suitable. They clearly identify the interest for this study and its possible relevance. They recap the information contained in the main text without repetitions.

Introduction: The Introduction provides adequate background. This section is concise, and includes specific literature references.

Materials and Methods: This section is clear, correctly and extensively presenting the study design, the methodology adopted and the subsequent evaluation of results.

Results: The results obtained by the authors are logically presented and accompanied by clear tables and figures. They are extensively described and substantially commented in the light of the aim of the study.  

Discussion: The scientific data presented are pertinent to the aim of the study. The discussion of data is well organized and balanced. The authors critically examine the results of data achieved in the light of the state of science highlighted in the introduction and the comments reported in discussion are pertinent. Discussion follows a logical line. The discussion of results is also extensive and clear.  The conclusions are drawn from the data related to the aim of the study. The paper offers the perspective for further study.

References: They are appropriate and present a good up-to-date of items on the argument.

Decision: The current manuscript is acceptable for publication after minor revision

Authors: Thank you. The manuscript was improved with your suggestions.
